# Measuring Time-Varying Effective Reproduction Numbers for COVID-19 and Their Relationship with Movement Control Order in Malaysia

**DOI:** 10.3390/ijerph18063273

**Published:** 2021-03-22

**Authors:** Kamarul Imran Musa, Wan Nor Arifin, Mohd Hafiz Mohd, Mohammad Subhi Jamiluddin, Noor Atinah Ahmad, Xin Wee Chen, Tengku Muhammad Hanis, Awang Bulgiba

**Affiliations:** 1School of Medical Sciences, Universiti Sains Malaysia, USM, Kubang Kerian, Kelantan 16150, Malaysia; wnarifin@usm.my (W.N.A.); tengkuhanismokhtar@gmail.com (T.M.H.); 2School of Mathematical Sciences, Universiti Sains Malaysia, USM, Penang 11800, Malaysia; mohdhafizmohd@usm.my (M.H.M.); subhi@student.usm.my (M.S.J.); nooratinah@usm.my (N.A.A.); 3Faculty of Medicine, Sungai Buloh Campus, Universiti Teknologi MARA, Sungai Buloh, Selangor 47000, Malaysia; drchenxw@uitm.edu.my; 4Centre for Epidemiology and Evidence-Based Practice, Department of Social and Preventive Medicine, Faculty of Medicine, University of Malaya, Kuala Lumpur 50603, Malaysia; awang@um.edu.my

**Keywords:** reproduction number, COVID-19, movement control order, epidemic curve

## Abstract

To curb the spread of SARS-CoV-2 virus (COVID-19) in Malaysia, the government imposed a nationwide movement control order (MCO) from 18 March 2020 to 3 May 2020. It was enforced in four phases (i.e., MCO 1, MCO 2, MCO 3 and MCO 4). In this paper, we propose an initiative to assess the impact of MCO by using time-varying reproduction number (*Rt*). We used data from the Johns Hopkins University Centre for Systems Science and Engineering Coronavirus repository. Day 1 was taken from the first assumed local transmission of COVID-19. We estimated *Rt* by using the **EpiEstim** package and plotted the epidemic curve and *Rt*. Then, we extracted the mean *Rt* at day 1, day 5 and day 10 for all MCO phases and compared the differences. The *Rt* values peaked around day 43, which was shortly before the start of MCO 1. The means for *Rt* at day 1, day 5, and day 10 for all MCOs ranged between 0.665 and 1.147. The average *Rt* gradually decreased in MCO 1 and MCO 2. Although spikes in the number of confirmed cases were observed when restrictions were gradually relaxed in the later MCO phases, the situation remained under control with *Rt* values being stabilised to below unity level (*Rt* value less than one).

## 1. Introduction

### 1.1. Movement Control Orders (MCOs) in Malaysia

To combat the coronavirus disease 2019 (COVID-19) pandemic, the Malaysian government initiated MCOs, effective on 18 March 2020, with the aim to aggressively contain the outbreak and slow down the transmission rate. This strategy was proclaimed under the Prevention and Control of Infectious Diseases Act 1988 (Act 342) and the Police Act 1967. Offenders violating MCO directives could face a maximum of RM 1000 fine and/or six months in jail or both if convicted [1]. A seven-week MCO was employed, containing four phases, followed by a five-week conditional MCO (CMCO) and subsequently, recovery MCO (RMCO) until to date [2,3,4]. The imposition of different MCO levels was decided by the Malaysian government, considering the livelihoods in communities and the economic stability.

Various non-pharmacological interventions had been implemented during MCO 1 that were continued in MCO 2; a gradual relaxation of rules occurred in subsequent MCO phases (Table 1). During MCO 3, a relaxation of MCO directives was imposed where some sectors could operate (e.g., certain construction projects and the automotive industry). However, the application was subjected for approval by the International Trade and Industry Ministry to reduce the number of employees in the workplace. Further expansion was allowed for almost all economic sectors to run at full capacity during MCO 4 [3,5]. Another relaxation of movement restrictions was implemented in which people could go out in pairs and beyond 10 km for daily essentials and medical needs, but they must be from the same household or otherwise with a valid reason [2,4].

On top of the four MCO phases that were implemented nationwide (Figure 1), the Malaysian government also used a more targeted approach in the form of enhanced MCO (EMCO). EMCO was imposed in areas where large clusters of positive cases were detected. The main objective of EMCO was to facilitate faster active case detection, screening, testing and isolation of infected high-risk populations. Residents in the affected areas were mostly confined to their homes where their food and daily supplies were delivered to. Non-residents and visitors from outside EMCO areas were also forbidden from entering the areas. Stricter measures were needed to reduce interactions between high-risk group and the “naïve” populations [1,3,4,6]

### 1.2. Reproduction Number (Rt) and COVID-19

With the increasing number of cases and the introduction of multiple non-pharmaceutical interventions (NPIs) by most countries, assessing and monitoring the transmissibility of the disease as a measurement of the effectiveness of control measures is necessary. Time-varying *Rt*, defined as the average number of secondary cases from a partially susceptible population per infectious case [7], is a universally applied indicator for assessing and monitoring the effectiveness of control measures.

The basic reproductive number before mitigation starts is named *R0*, whereas the reproduction number after mitigation starts is named *Rt*; both can be used to measure the transmissibility of the infection. A value of *Rt* which is more than 1 means that the infection is spreading, with additional new cases being generated at an exponential rate. A value of *Rt* which is less than 1 means that the spread of infection is decreasing. Theoretically, we need information about generation time, which is defined as the period between the infection of the index and the next case. However, this information is usually difficult to ascertain. Therefore, information regarding the serial interval (defined as the interval between disease onset in the index and the next case) distribution in the data is used instead [7,8,9,10,11].

A recent systematic review reported that the basic reproduction (*R0*) for COVID-19 was 3.38 (standard deviation (SD) = 1.40) and a range of 1.90–6.49 [12]. An early report from Wuhan showed that the *R0* was estimated to be 2.2 (95% CI, 1.4–3.9) [9] or higher at 4.08 [10].

Several studies have indicated the feasibility of using *Rt* to assess and explain transmissibility dynamics and epidemic progression [11,12,13]. *Rt* reflects, although with limitation, the spread of infection. In the case of COVID-19, NPI has been thought to reduce the spread through isolation and contact tracing. Both reduce the time during which cases are infectious in the community, thereby reducing *Rt* [8,13,14]. If control efforts bring *Rt* below 1, then on average, the number of new cases reported can decline, which can become apparent after a delay of approximately one incubation period plus time to case detection and reporting, following the implementation of the control measure (i.e., at least two weeks) [14].

The relationship between MCO and *Rt* may indicate the impact of MCO. The numbers can be quantified to indicate the effectiveness of the impact of MCO on the spread of COVID-19. This indication, in return, can provide critical information to help epidemiologists and public health workers strategize their COVID-19 control programme.

The general objective of the study is to use the daily incidence data for confirmed COVID-19 cases in Malaysia to quantify time-varying effective *Rt*. Specifically, using the incidence number of confirmed COVID-19 cases between 4 February 2020 (assumed day 1 of local COVID-19 cases) until 16 May 2020 (day 102), we (a) measure the *Rt* of SARS-CoV-2 at day 1, day 5 and day 10 after the initiation of each of the four MCOs and (b) quantify the impact of each MCO by measuring the difference in the *Rt* between the beginning of MCO (day 1) and day 5 and between day 5 and day 10 of each MCO.

## 2. Materials and Methods

### 2.1. Source of COVID-19 Data and Variables

We downloaded the COVID-19 data by using the R software from this link: https://github.com/RamiKrispin/coronavirus/tree/master/csv (accessed on 10 September 2020). The owner of this GitHub repository compiled the COVID-19 data retrieved from the Johns Hopkins University Centre for Systems Science and Engineering (JHU CCSE) coronavirus repository (https://github.com/CSSEGISandData/COVID-19) (accessed on 10 September 2020).

The downloaded data are in the time series format and contain variable date, province, country, latitude, longitude, type and case. For the analysis, we selected observations to only include data from Malaysia and type equals “confirmed,” which refers to confirmed COVID-19 cases. The variable case is the variable that contains the number of cases reported every day.

We considered the first day of our data to be 4 February 2020, because we assumed that was the earliest evidence of local COVID-19 transmission as reported by the Ministry of Health (MOH), Malaysia. We included data up to 16 May 2020, that is, day 102.

The study did not require ethical approval because we used data that are publicly available at https://github.com/RamiKrispin/coronavirus/tree/master/csv (accessed on 10 September 2020) and are in an aggregated format. The original source of data (raw data) came from the JHU CCSE coronavirus repository (https://github.com/CSSEGISandData/COVID-19) (accessed on 10 September 2020).

### 2.2. MCOs

The duration of different MCO levels (Figure 1) was decided by the Malaysian government. Six MCOs were implemented: (a) MCO 1, MCO 2, MCO 3 and MCO 4, (b) CMCO and (c) RMCO. In this study, the data were analysed for the first four MCO phases (the period from MCO 1 to MCO 4). The first three MCOs lasted for 14 days, and the last MCO (MCO 4) lasted for one week. For MCO 1, MCO 2 and MCO 3, we identified day 1, day 5 and day 10. As MCO 4 lasted only a week, day 10 was taken from the third day of the subsequent MCO (CMCO). 

### 2.3. Estimating Rt

Data were analysed using the R IDE software version 4.0.2 (CRAN, Vienna, Austria) [15]. We employed the EpiEstim package (CRAN, Vienna, Austria) in R IDE to quantify the infection transmissibility over time during an epidemic based on a Bayesian approach. The EpiEstim package (*estimate-R* function) allows us to estimate the instantaneous and case *Rt* for COVID-19 by using (a) a time series of COVID-19 incidence and (b) the distribution of the COVID-19 serial interval (time between symptoms onset in a primary case and symptoms onset in the secondary case of COVID-19) [12,13].

In the *estimate-R* function, for estimating the Malaysia COVID-19 instantaneous or time-varying *Rt*, we initially used the values of a mean of 7.5 days and SD of 3.4 days to represent the serial intervals (SIs) [9]. Subsequently, we used the mean SI value of 3.96 days and SD of 4.75 [10] to estimate the next *Rt*. We assumed for our analysis that the COVID-19 SI follows a normal distribution. The EpiEstim package uses the Poisson likelihood to calculate the instantaneous *Rt* by using the sliding window of seven days [8,12,13]. The results from the EpiEstim analysis include a) the plot of the epidemic curve for COVID-19 incidence and b) the plot of *Rt* on sliding weekly windows. We uploaded the complete *R* codes for the analyses at https://github.com/drkamarul/R0_MYS/blob/master/repro_covid19_malaysia.md (accessed on 10 September 2020). 

We also extracted the mean *Rt* values at day 1, day 5 and day 10 for each of the four MCOs (the *Rt* at day 10 in MCO 4 was taken the third day of CMCO). To see the trend of the mean *Rt* over 10 days for each MCO, we calculated the difference of the mean *Rt* between (a) day 5 and day 1 and (b) day 10 and day 5; we also plotted the *Rt* values (the mean and 95% credible intervals) by using the line plots.

## 3. Results

We analysed the daily Malaysia COVID-19 incidence data starting from 4 February 2020 (day 1) to 17 May 2020 (day 102). The incidence data reported by the MOH, Malaysia refer to the day when a patient was detected positive by the RT-PCR test. In the data, the variable date corresponds to the date that the test became positive and not the onset of the symptoms. All COVID-19 tests in Malaysia are performed by the COVID-19 gazetted laboratories using the RT-PCR antigen test.

Figure 2 shows the epidemic curve based on the daily confirmed COVID-19 cases reported by the MOH, Malaysia. The growth of COVID-19 spread took place between day 23 after the first local case of COVID-19 (4 February 2020) and day 43. From day 40 to day 60, the growth peaked. Overall, this observation agrees with the number of reported active COVID-19 cases in Malaysia, which peaked around 5–8 April 2020, with the highest 2596 active cases recorded on 5 April [16]. From then onwards, the number of active cases started to decrease and, from day 70 to day 100 (14 May 2020), the number of new daily COVID-19 cases slowly decreased and plateaued.

The transmissibility of COVID-19 in Malaysia was estimated on the basis of *Rt* from day 1 (4 February 2020) to day 100 (14 May 2020). The plot of *Rt* in Figure 3 is based on the serial interval with a mean of 7.5 days and SD of 3.4 days, whereas the plot of *Rt* in Figure 4 is based on the serial interval with a mean of 3.96 days and SD of 4.75 days. Both plots share similar characteristics; a peak between day 28 and day 32 and a small peak at day 43. The *Rt* values decreased gradually after day 43, remained above 1 until day 65 and after which reduced to below 1. In comparison with the epidemic curve in Figure 2, we can associate the first peak with the initial surge of confirmed cases between day 28 and day 32. However, at this early stage of the epidemic, the data population was still very low and might not be enough to capture the essential features of the epidemic. These observations were supported by the wide 95% percentiles for values observed on day 1 to day 45, which is the most likely reason for the significant difference in the profiles of the first peak (in Figure 3 and Figure 4), as we changed the parameter values (mean and SD). The second peak at day 43 corresponded to another spike of confirmed cases from day 39 onwards, which also saw the beginning of the second wave of COVID-19 infections in Malaysia. Timing wise, the second peak coincided well with the events that occurred 14 days prior to that which was the religious gathering (the Tabligh group) in Sri Petaling, Kuala Lumpur, Malaysia. More than 16,000 attendees of the Tabligh group gathered in this annual occasion, which took place between 27 February 2020 and 1 March 2020 [17,18].

The decrease in *Rt* values soon after the second peak indicates reduced transmissibility. With *Rt* values greater than 1, the number of confirmed cases continued to increase, as illustrated in Figure 2. However, the reduced transmissibility slowed down the rate of increase. We can observe this phenomenon from Figure 2, where the average number of confirmed cases from day 43 to day 65 remained between 100 and 200 cases daily. This observation evidently shows that the MCO, which started at day 44, had a direct role in this turn of events. Strict measures taken in the first phase of the MCO managed to tame the exponential increase in infection. In Figure 3 and Figure 4, the reduction in *Rt* values appeared to start before the imposition of MCO. The sensitivity of the *Rt* values to the choice of parameter values (i.e., mean and SD) might have introduced some level of uncertainty in the exact timing of the second peak. A further investigation is obviously needed to verify the accuracy of this part of the results.

Stricter measures imposed during MCO 1 had a significant impact on the fast reduction in *Rt* values from day 44 to day 58. The strict measures which continue during MCO 2 managed to bring down *Rt* values to below 1 after day 65. As *Rt* values dipped below 1, the number of confirmed cases in Figure 2 also began to drop.

*Rt* values remained below 1 for most of MCO 3, indicating that the transmission of the disease was under control. As the MCO entered phase 4 (MCO 4), the further relaxation of restrictions brought a few spikes in the number of cases. Nevertheless, the transmissibility remained well under control because the *Rt* values had already reduced significantly and hovered around 1 or less by then.

Using the *Rt* estimated from the second SI (mean = 3.96 days, SD = 4.75 days), instantaneous mean *Rt* values for day 1, day 5 and day 10 for each MCO phase are calculated based on seven-day rolling average and the results are illustrated in Figure 5. Detailed numerical information is compiled in Table 2, where “day started” and “day ended” correspond to the period for the calculation of the seven-day rolling average. The percentiles are reported for the 2.5 and 97.5 percentiles of the *Rt* values.

In this illustration, the connection between the different phases of MCO and *Rt* values is elucidated further. From day 1 to day 10 of MCO 1, a progressive drop in the instantaneous mean is observed and subsequently, from day 1 to day 10 of MCO 2 a similar monotonic decrease persists. It is important to bear in mind that during MCO 1 and MCO 2, the entire country was under an almost complete lock down with very restricted movement. The strict measures are reflected in the continuous decrease in the instantaneous mean *Rt* values and by day 5 of MCO 2, the value had decreased to below the threshold value of 1. In this case, we can see that the instantaneous mean *Rt* values provide a stronger indicator of the impact of strict MCO that manage to bring down the mean *Rt* value from 1.23 to below 1 in about two weeks.

During the first 10 days of MCO 3, a steady increase in the instantaneous mean *Rt* values is detected. However, the rise in values do not persist in MCO 4 where we observe that between day 1 and day 10 in MCO 4, the instantaneous mean *Rt* declines and arrives at fairly steady values (from 0.8 to 0.9). At this point, it is unclear whether the rise in values is due to the relaxation of MCO directives during MCO 3 which saw some sectors were allowed to operate (e.g., certain construction projects and automotive industry). Upon scrutiny of the timeline of COVID-19 in Malaysia, we learned that a large cluster was detected during this period which involves students returning from Temboro, Magetan, Indonesia with 43 cases (day 5, MCO 3) and later 72 cases from the same cluster (day 1, MCO 4) [15]. This cluster was obviously not a contributing factor to the community transmission during MCO 3. Therefore, it does affect the computed values of *Rt*, but the effect is not significant enough to influence the long-term behaviour. The fact that the instantaneous mean *Rt* declines soon after and the 95% percentile remains mostly below the threshold value of 1 throughout MCO 3 and MCO 4 goes to show that the pandemic has been tamed sufficiently.

## 4. Discussion

The MCO enforced by the Malaysian government is an example of an NPI, which is the intervention proposed strongly by epidemiologists and infectious disease modellers to halt the transmissibility of SARS-CoV-2; NPI aims to prevent and/or control SARS-CoV-2 transmission in the community [14,19]. Until a safe and effective vaccine is available to all those at risk of the severe COVID-19 disease, NPI will continue to be the main public health tool against SARS-CoV-2. Intensive NPIs reduce the isolation delay period. The isolation of an infector one day earlier is expected to reduce the mean serial interval by 0.7 days and thus can lead to the shortening of the serial interval by more than three days if the infector is rapidly isolated [13]. In general, countries that enforced movement control or lockdown experienced a reduction in *Rt* [8,13,14,20].

The Malaysian government initiated MCOs from 18 March 2020 to 3 May 2020. It had three types: MCO, CMCO and RMCO. The MCO was also implemented in four phases (i.e., MCO 1, MCO 2, MCO 3 and MCO 4) nationwide, with the main objective to stop or curb the spread of the SARS-CoV-2 virus. In order to quantify the impact of MCO during the four different phases, we have analysed the daily incidence data to compute time-varying effective *Rt* values as well as the instantaneous mean *Rt* values on day 1, day 5 and day 10 after the initiation of each of the MCO phases.

### Main Results

The main results can be summarized as follows:**The overall trend of *Rt* values do correspond with the timeline of the pandemic and the level of strict measures imposed during each phase of the MCO**. The *Rt* values peaked around day 43, which was shortly before the start of MCO 1. A marked decline in *Rt* values was observed during MCO 1, and the reduction in transmissibility is reflected in a reduction in the rate of increase in daily cases. Thus, we can say that measures taken during MCO 1 were successful in taming the exponential growth of the number of infections. The *Rt* values started to fall below the threshold value of 1 during MCO 2, at which time the number of daily cases began to show a downward trend. *Rt* values remained below 1 throughout MCO 3. The relaxation of certain restrictions during MCO 3 and MCO 4 might have allowed a few spikes of confirmed cases and this is indicated by a slight increase in *Rt* values, but the *Rt* values remained stable and hovered around 1 and slightly below that.**The instantaneous mean *Rt* values computed for day 1, day 5 and day 10 of each phase of the MCO provide further insight into the suitability of *Rt* values as a quantifiable indicator to the impact of each MCO phase**. From day 1 to day 10 of MCO 1, the instantaneous mean *Rt* reduces progressively and monotonically and this behaviour persists until day 10 of MCO 2. The continuous decrease in the mean *Rt* is a reflection of the strict measures imposed during MCO 1 and MCO 2. The impact of strict MCOs managed to bring down the mean *Rt* value from 1.23 to below 1 in about three weeks where at this level of *Rt*, the spread of infection is expected to be under control. The relaxation of the level of restrictions during MCO 3 and MCO 4 is also reflected in the instantaneous mean *Rt*, where a steady rise of the values is picked during MCO 3.**Sensitivity of *Rt* values with respect to the choice of parameter values.** Our results reveal the sensitivity of *Rt* values with respect to the choice of parameter values, namely, the mean and SD. This is particularly a concern, especially at the start of the pandemic where two significant peaks of *Rt* values were observed. A change in mean and SD values modified the profile of these peaks rather abruptly. Our hypothesis is that the sensitivity is due to the small sample size problem where data are insufficient to capture the essential features of the dynamical system. This hypothesis is supported further by the wider 95% percentiles for values observed on day 1 until day 45. Although reasons exist to believe that these peaks represent the relevant historical events of the pandemic (peak 1 can be associated with the start of the second wave of the pandemic in Malaysia that started around 27 February 2020 (day 23), whereas peak 2 can be tied up with the wave of infection originated from a mass gathering at Sri Petaling, Kuala Lumpur, which took place between 27 February 2020 and 1 March 2020), the accuracy of computed *Rt* values and the time they occurred should be investigated further.**Sensitivity to outliers.** In the presence of outliers, the instantaneous mean *Rt* values can be affected. This is observed in our analyses of the mean values during MCO 3 and MCO 4 and the associated timeline of the pandemic in Malaysia. The outlier is a cluster of cases involving students returning from Temboro, Magetan, Indonesia which is not a contributing factor to the community transmission during MCO 3. During the first 10 days of MCO 3, a steady increase in the instantaneous mean *Rt* values was detected, peaked around the first day of MCO 4, and then decreased to steady values between 0.8 and 0.9. It is possible that the rise of cases may also be due to the relaxation of certain measures during MCO 3, however, the rising trend does not persist in MCO 4. This gives the indication that the effects of outliers on *Rt* are possibly short-lived and do not contribute significantly to the baseline value associated with community transmission that is directly impacted by MCO measures. This supposition is further strengthened by the fact that the 95% percentile of the instantaneous mean *Rt* remain mostly below the threshold value of 1 throughout MCO 3 and MCO 4.

We have shown in our analyses of the *Rt* values and a close scrutiny of the timeline of COVID-19 in Malaysia that the impact of the MCO phases is quantifiable by the *Rt* values. From our *Rt* calculation, we are able to highlight the importance of the strict measures imposed in MCO 1 and MCO 2 in bringing the *Rt* values down to below the threshold value of 1 in a matter of weeks. The continuous steady decrease in *Rt* values is indicative of the role of MCO 1 and MCO 2 in slowing down the spread of the pandemic. This factor is crucial for developing countries, such as Malaysia, which face problems of limited medical resources (e.g., tests, drugs and hospital beds), to ensure that the surge in the number of people requiring medical care due to COVID-19 at any particular time is reduced, and the healthcare system does not exceed its capacity.

Another useful discovery from our analyses is the importance of making sure that *Rt* values have reached below the threshold value of 1 before introducing any relaxation of MCO measures. Once certain restrictions are lifted, there are bound to be spikes in the number of cases which will cause *Rt* values to increase. However, if *Rt* values have reached below 1, the situation can still be kept under control.

Our results do provide an indication of the success of the NPI enforced by the Malaysian government during the period between 18 March 2020 to 3 May 2020. However, we would like to express caution that the insights from our analysis also depend on data that come from the MOH, Malaysia that include laboratory testing and contact tracing strategies. For instance, the number of confirmed cases can often be determined by the speed of the notifications of COVID-19 tests, the accuracy of the test and the maximum number of COVID-19 tests performed daily by the MOH. In simple words, with additional tests conducted, new cases may be recorded. Still, the quality of the data can be better than before, and the predictability of the pandemic using quantities such as *Rt* can be further improved.

Timely track and trace process is undoubtedly one of the strategies in curbing the pandemic, and Malaysia’s MySejahtera apps play a vital role, which has reached almost 60% of the Malaysian population [21]. Given the limitations of the app in performing optimum contact tracing (e.g., low number of elderlies using the apps, lack of Bluetooth-enabled proximity sensing), the Ministry of Health (MOH) Malaysia implemented both manual contact tracing (the conventional method frequently used in managing other infectious diseases in Malaysia such as dengue and tuberculosis) and app-based detection simultaneously. This approach of “track and trace” has remained unchanged and constant until beginning of the year 2021 [22], hence it is believed that the effect of tracing/testing delay on our modelling is minimal [23]. A report published in June 2020 presented that a robust contact tracing team is one of the factors contributing to the success of containing the infection in the early phase of the COVID-19 outbreak in Malaysia [24].

On 2 March 2020, the estimated COVID-19 test for Malaysia was less than 0.01 per 1000 people (while for South Korea it was 0.23 per 1000 people), and it gradually increased to 0.28 per 1000 people on 23 May 2020 (0.21 per 1000 people for South Korea) [25]. Though the per capita test seems to be low, there were on average 29.7 tests per confirmed COVID-19 case. This is in line with recommendation by the World Health Organization; 10 to 30 tests per confirmed case as a general benchmark. The low per capita test may falsely reduce the value of *Rt*. In our analysis, we believe the effect of the low per capita test on our *Rt* calculation is negligible because *Rt* is more dependent on the trend of confirmed COVID-19. Our analysis assumes that there was not much individual variation in the COVID-19 infection in Malaysia despite a few COVID-19 outbreaks in South Korea, USA and Poland, indicative of the presence of super-spreaders. The variation of individual infection is measured by a metric, known as K (a low K value suggests that a small number of infected people are responsible for large amounts of disease transmission). The presence of a super-spreader will significantly increase the size of the COVID-19 cluster in a short time interval. The K value is more are critical in the late stages of the epidemic when the virus is almost eradicated, unlike the focus of this paper, where the interest was on the early part of the epidemic [26,27].

The main source of limitation is using the date of a notification of a positive COVID-19 test to construct the epidemic curve and to estimate *Rt*. The ideal parameter required to estimate *Rt* is the generation time. However, in most studies, the generation time is impossible to ascertain. The next best parameter is the SI of the infection. Unfortunately, we do not have data about infectors and infectees to estimate the SI. A few studies have shown that using laboratory notifications can still provide a useful reference to the transmissibility of the SAR-Cov-2.

The *Rt* estimates reflect the average value for the country. However, different states and even different districts within a state may likely have different *Rt* values because the response to MCOs, the population density and the bulk of infections vary. This study utilises the number of confirmed cases, although the exact number of infected cases can be larger if asymptomatic cases are considered. We estimate that 5–80% of cases are asymptomatic.

## 5. Conclusions

In Malaysia, the daily confirmed case data indicate that the epidemic curve for COVID-19 reached a peak between day 40 and day 70. However, the time-varying *Rt* peaked earlier (between day 25 and day 35) which is before the start of MCO and thus may reflect the response from the community and the government. Strict MCO 1 and MCO 2 correspond to a gradual decrease in *Rt*, but relaxed MCO 3 and MCO 4 correspond to a slight increase in *Rt* before it plateaued below 1.

## Figures and Tables

**Figure 1 ijerph-18-03273-f001:**
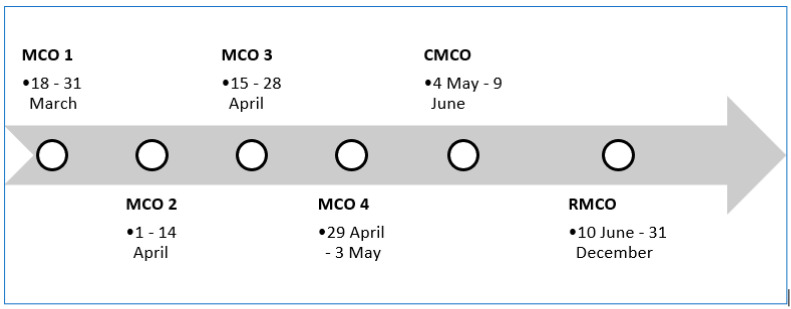
Six MCO phases in Malaysia. MCO: movement control order, CMCO: conditional movement control order, RMCO: recovery movement control order. Source: [1,3,4,6].

**Figure 2 ijerph-18-03273-f002:**
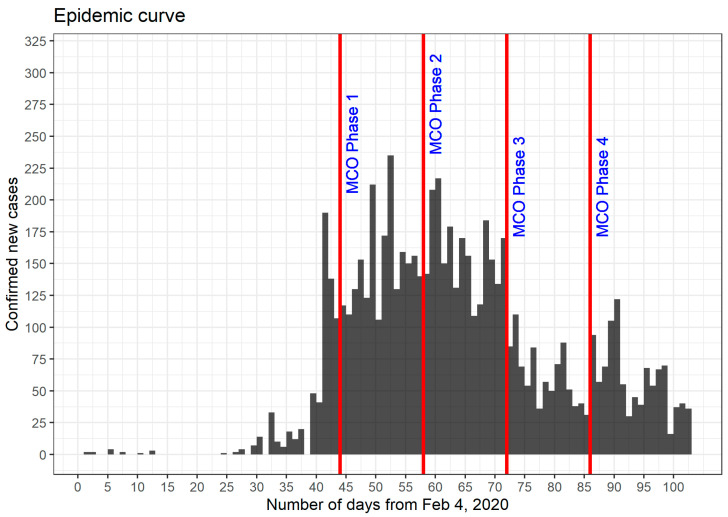
Epidemic curve between day 1 (4 February 2020) and day 102 (16 May 2020) for Malaysia.

**Figure 3 ijerph-18-03273-f003:**
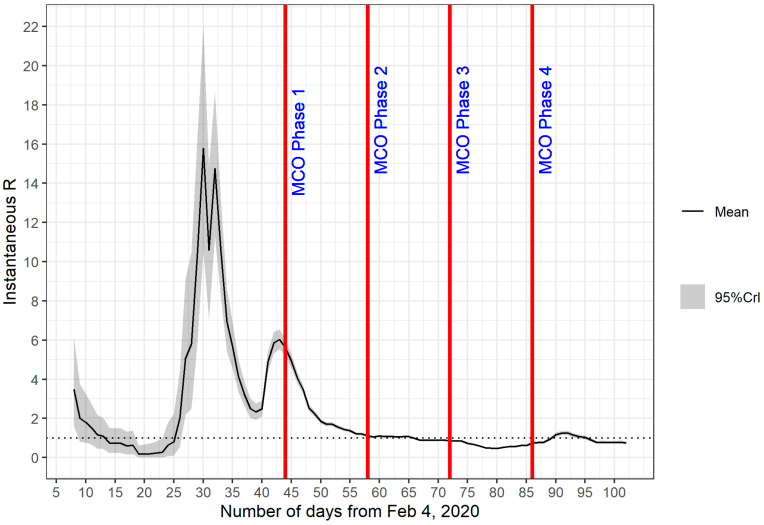
*Rt* from day 1 (4 February 2020) to day 102 (16 May 2020) on sliding weekly windows. The estimate was based on the serial interval with mean = 7.5 days and SD = 3.4 days.

**Figure 4 ijerph-18-03273-f004:**
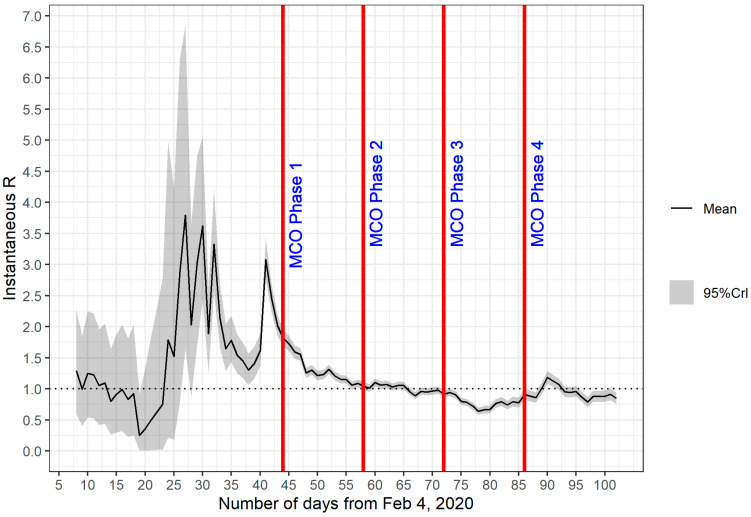
*Rt* from day 1 (4 February 2020) to day 102 (16 May 2020) on sliding weekly windows. The estimate was based on the serial interval with mean = 3.96 days and SD = 4.75 days.

**Figure 5 ijerph-18-03273-f005:**
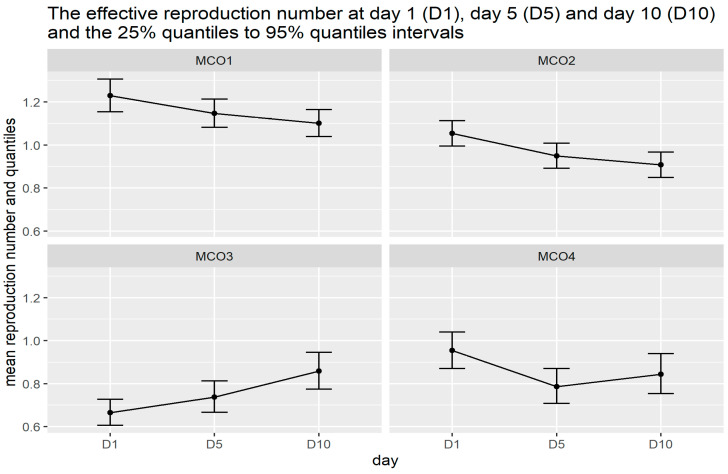
*Rt* values and their 25% and 95% quartile intervals at D1, D5 and D10 for each MCO phase.

**Table 1 ijerph-18-03273-t001:** Non-pharmacological interventions (NPIs): movement control orders (MCOs) in Malaysia. CMCO: conditional MCO; RMCO: recovery MCO.

Phase	NPIs
**MCO**	**Movement control of the public (i–viii) and border control (ix–xii)** Prohibition of mass gatherings for religious, sports, social and cultural activities. Closure of all places of worship, including mosques, churches and temples. Closure of all business premises, except for supermarkets, public markets, grocery stores and convenience stores. Closure of all educational institutions, including pre-schools, government and private schools (e.g., daily schools, Tahfiz centres, international schools) and higher education institutions. Closure of all government and private sector premises, except for those providing essential services (e.g., water and electrical services, telecommunication services, transportation (air/land/sea), banking services, healthcare and medical services, fire and rescue, prisons, defence and security, cleaning, retail and food supply services). Food premises are prohibited to provide dine-in services but take-away and food delivery services are allowed. Limited operation hours of the public transport. Only one person per household can leave home for daily necessities and medical care, unless the accompaniment is reasonably necessary, and the travel is subjected to a radius of 10 km. Suspension of Malaysians travelling abroad. Mandatory medical examinations upon arrival and a 14-day quarantine for all entries into Malaysia. Prohibition of interstate travel unless a written police permit with a valid reason is obtained. Restrictions on the entry of all tourists and foreign visitors into the country.
**CMCO**	Closure of Malaysia borders and prohibition of international travelling. Relaxation of restrictions to most economic sectors, with business standard operating procedures (SOPs), including physical distancing, temperature checks, recording the names and contacts of customers. Closure of pubs, theme parks, cinemas and entertainment centres and prohibition of conferences and exhibitions. Suspension of sports activities involving mass gatherings, body contact, indoor and stadium sports events. Closure of all schools and education institutes. Restriction of interstate travel, except for work purposes and return to workplace/home after being stranded in hometowns. Prohibition of mass gatherings, including religious, social and cultural activities.
**RMCO**	Closure of Malaysia borders and prohibition of international travelling, except for specific categories of foreigners to enter Malaysia (e.g., foreign diplomats and ambassadors, members of Malaysia My Second Home Programme, foreigners under the medical tourism industry and expatriates with working visas). Mandatory health inspection and a 14-day quarantine for all entries into Malaysia. All foreigners will bear the full cost of quarantine services. Interstate travel is allowed, except for areas under EMCO. Closure of night pubs and theme parks and prohibition of mass religious activities and social gatherings. Suspension of sports and games involving mass gatherings of supporters in stadiums. Mandatory face mask-wearing in public places from 1 August 2020.

**Table 2 ijerph-18-03273-t002:** Mean value of *Rt* at day 1, day 5 and day 10 for each MCO in Malaysia.

Day	MCO	Mean *Rt*	Difference	Difference (%)	Day Started	Day Ended	2.5 Percentile	97.5 Percentile
D1	1	1.23	REF	REF	45	51	1.155	1.307
D5		1.147	−0.082	−6.69	49	55	1.082	1.214
D10		1.101	−0.046	−4.023	54	60	1.039	1.165
D1	2	1.054	REF	REF	59	65	0.995	1.114
D5		0.949	−0.104	−9.913	63	69	0.892	1.008
D10		0.907	−0.042	−4.4	68	74	0.849	0.967
D1	3	0.665	REF	REF	73	79	0.606	0.728
D5		0.738	0.072	10.869	77	83	0.667	0.813
D10		0.858	0.12	16.283	82	88	0.774	0.946
D1	4	0.954	REF	REF	87	93	0.871	1.041
D5		0.787	−0.167	−17.526	91	97	0.707	0.87
D10		0.844	0.058	7.317	96	102	0.754	0.939

**Day**: Day 1 = D1, Day 5 = D5, Day 10 = D10, REF = REFERENCE VALUE.

## Data Availability

The dataset can be downloaded from https://github.com/RamiKrispin/coronavirus. Direct download of the data is available at https://raw.githubusercontent.com/RamiKrispin/coronavirus/master/csv/coronavirus.csv. The analysis and *R* codes are available at https://github.com/drkamarul/R0_MYS/blob/master/repro_covid19_malaysia_06112020.md.

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
