# Peer review of "Measuring Time-Varying Effective Reproduction Numbers for COVID-19 and Their Relationship with Movement Control Order in Malaysia"

_ijerph, 2021, doi:10.3390/ijerph18063273_

Round 1

Reviewer 1 Report

Major revised

Reviewer 2 Report

This is important study to quantify the time-varying reproduction number (Rt) for Malaysia using the COVID-19 incidence data and quantify the impact of each Malaysian's Movement Control Order (MCO) by measuring the difference in the Rt between pre and post of the MCO. The paper is interesting. However, I have few comments for authors consideration.

1. In order to perform robust analyses of transmission in the early stages of this outbreak. I think it is important to account for two distinct populations in the case notification data — imported and locally acquired. For example authors may consider to estimate time-varying R value based on cases that have been identified as a result of local transmission, whereas imported cases contribute to transmission only. Alternatively authors may consider that the method assumes that local and imported cases contribute equally to transmission or likely that imported cases contributed relatively less to transmission than locally acquired cases. Hence, different type of scenario may be good to be explored in their simulation estimations.

2. According to the authors, the main source of limitation is the use of the date of notification of a positive COVID-19 test to construct the epidemic curve and to estimate the Rt. I was wondering if authors able to elaborate information on time delays between illness onset and case notification using their local sources. It would be good if authors can extract this information from their Ministry of Health since incorporation of this lag is critical for accurate interpretation of the most recent data in the analysis, to ensure that an observed drop in the number of reported cases represents an actual drop in case numbers.

3. Lastly, I think it is important to discuss the implication of the findings from Rt estimates and relative impact of specific non-pharmaceutical interventions (i.e  MCO, CMCO etc) on the spread of the COVID-19 for the next phase of response planning by decision makers and some challenging issues. 

Reviewer 3 Report

This study aimed to measure the time-varying effective reproduction (Rt) numbers for COVID-19 with movement control order (MCO) in Malaysia, and provided some indicator for assessing the effectiveness of those control measures. There are some interesting results in this manuscript, but there are some major and minor concerns to improve it. 1, Abstract should be reorganized. Authors should write the main findings and conclusion, but not the detailed methods. 2, Their data showed that the spread of SAR-Cov-2 started to decline even before MCO, and authors should give some explanations for this observation. 3, The time-point is confused. For example, figure 1 showed that the intervals of MCO4 was April 29-May 3 ( only Five days), but the intervals in Abstract is (Day 1 to Day 10 MCO 4)? Why they split those data into four intervals, and 10 days for each interval? 4, They estimated the R0 and Rt based on the number of reported COVID-19 cases from JHU CCSE. This is not true cases, for example, there are many infected but asymptomatic patients. Authors showed mention this. 5, Some use of English should be more professional.

Reviewer 4 Report

The manuscript by Musa et al. use data form Malaysia to estimate Rt, and correlate the time course of Rt with movement restriction.

The aim of the paper is clearly written, and - though the paper does not estend the available modeling techniques, or thequniques to analyze data - it adds to our knowledge how contact restrictions effect the spread of an infectious diseases. In that, the paper is a valuable contribution to the existing literature.

However, I would like to ask the authors to clarify some questions.

1) There are 4 MCO's (MCO1-MCO4). It is not clear to me, if they did differ, or if they are merely extensions of the very same movement control rules. If the implemented rules in MCO1-MCO4 are different, please specify (you mention in lines 255-262 that MCO1+2 have been stricter than MCO3+4; for the reader, it is hard to follow in which respect that has been the case). If they are identical, why do you distinguish between them?

2) lines 126-130:
You consider different time intervals (roughly 5 days). It is not clear to me, where you require theses intervals. In your central figures (fig.3+4) you seem to work on a daily basis. Perhaps you want to explain the reader where these small intervals enter the analysis and how the analysis is affected by this choice.

3) Lines 137-139
Can you explain why you choose for a normal distribution for the serial interval? It is well known that the serial interval is strongly affected by the choice of this distribution (. J. Wallinga, M. Lipsitch, How generation intervals shape the relationship between growth rates and reproductive numbers. Proc. R. Soc. B 274, 599–604 (2007)).

I expect that the serial interval for SARS-CoV-2 is discussed in literature (see, e.g., https://doi.org/10.1093/jtm/taaa115 or DOI: 10.1126/science.abc9004 ). A more in-depth analysis of that aspect would perhaps improve the results. There might be differences from country to country, but these differences are most likely not that large.

4) Fig. 3:
The large R_t at day 25 is most likely an artifact, which is (again most likely) why you use a different serial interval for Fig. 4. I'm not sure why you keep both figures. In any case, it would be nice if you could discuss the aspects that might induce this large jump in Rt. There are several points that perhaps could influence this effect: (a) at that time, only are few cases. Which does mean that stochasticity might have a large influence. (b) We have asymptomatic cases, it is not clear how they affect the estimation in that early state (later, if we have many cases, the estimation is most likely not or only to a minor extend affected) (c) Awareness and more testing might lead to the diagnose  very of cases with only mild symptoms, which have overlooked before.

5) line 199:
You use a normal distribution with mu=3.69<sigma=4.75. In that case, there is a considerable part of the normal distribution in negative vales – true? How did you handle that?

6) Particularly, the discussion of asymptomatic cases is missing throughout the paper - asymptomatic cases form a major difficulty in most data analysis. As indicated above, that could also be the case for the present study

Round 2

Reviewer 3 Report

Authors have addressed some concerns, but there are still a lot of mistakes in their manuscript.

Abstract:“Rt values being stabilised to below unity”mean what?

Figure 3, Figure 4: what is Estimated R? Instantaneous R?

Figure 5: lack of statistical analysis; please delete the text above the picture.

Table 2: Please adjust the first line of the header to one line

Line 356: delete “4.1”  or delete this line.
